TECHNICAL RELEASE

# Inference of admixture in dogs from whole genome sequences

Gregory Kislik[1,*], Garrett Moore[2,†], Liudmilla Rubbi[1],
Veninka Nikki Supara[1], Grace Chen[1] and Matteo Pellegrini[1]

1 Molecular Cell and Developmental Biology, University of California, Los Angeles (UCLA), CA, Los Angeles, USA
2 Prosper DNA, Los Angeles, CA, USA

## ABSTRACT

**Background:** Understanding the genetic architecture of domestic dogs provides unique insights into the processes of domestication, breed formation, and the genetic basis of complex traits and diseases. Dog populations, characterized by their diverse morphologies and behaviors, also exhibit extensive evidence of historical and ongoing admixture. This widespread mixing, driven by both natural migration and selective breeding practices, has profoundly shaped the genomic landscape of modern dog breeds. Though global admixture has been extensively estimated in human population studies, where the number of subgroups is typically limited, there has been more limited analysis in canines, where there may be dozens of ancestral groups, or breeds.
**Results:** Here we present a procedure for estimating global admixture in dogs from whole genome sequence data using SCOPE. We created a reference population of 65 dog breeds that included 349 individuals, from which we determined breed-informative SNPs. We demonstrate that SCOPE can accurately infer breed composition in both simulated and real admixed samples, even at low sequencing depths. We also characterized the genetic similarity between our reference dog breeds and recovered previously reported relationships.
**Conclusion:** This approach allows us to identify the strength of the genetic signature of breeds and place error bounds on admixture estimates. It also provides evidence that admixture can be accurately inferred in subjects that may originate from multiple ancestral populations.

Submitted: 26 September 2025

\* Corresponding author. E-mail: gkislik@g.ucla.edu

† Current address: Waymo, Mountain View, CA, 94043, USA.

Preprint submitted at https://doi.org/10.64898/2026.02.09.704954

**Subjects** Genetics and Genomics, Animal Genetics, Bioinformatics

## INTRODUCTION

Recent genomic studies have revealed extensive admixture among domestic dog breeds as well as between dogs and wild canids. Large-scale analyses using genome-wide SNP data and whole-genome sequencing have shown that modern breeds often derive from multiple ancestral lineages, reflecting complex breed development histories shaped by geographic separation, selective breeding, and hybridization events [1, 2]. Deep-time analyses further indicate contributions to the gene pool of domestic dogs from at least two ancient wolf populations [3, 4]. In addition, studies of hybrid populations such as the Czechoslovakian wolfdog and Australian dingo demonstrate ongoing gene flow between wild and domestic canids, highlighting the dynamic nature of dog ancestry [5, 6]. Collectively, this body of work underscores the complex evolutionary history of domestic dogs and their close relatives.

Previous large-scale studies such as Parker *et al.* [1], analyzed over 160 breeds using ADMIXTURE and haplotype sharing to infer broad breed groupings and historical migration patterns. However, these studies do not focus on the accuracy of estimation of admixture in

individuals. Global ancestry, inferred by tools such as SCOPE [7] and ADMIXTURE [8], attempts to infer the proportions of an individual's genome that belong to an ancestral breed or group. Local ancestry, estimated by tools such as RFmix and Gnomix, instead attempts to assign blocks of loci to an ancestral population [9]. Local ancestry is a more constrained approach to admixture inference, as it assigns ancestry at each genomic locus to a discrete source population rather than estimating genome-wide ancestry proportions [10]. Beyond this, genetic drift can lead to differences between global and local admixture estimation because it randomly alters population allele frequencies [11]. Similarly, natural selection can remove loci important for local admixture estimation and cause ancestry inferences to differ substantially from genome-wide measurements [12].

Global admixture inference is computationally more efficient than local ancestry inference, as it estimates genome-wide ancestry proportions rather than assigning ancestry at each genomic locus. Various tools have been developed for rapid global admixture inference in large datasets. One of the most scalable and efficient is SCOPE, which utilizes a robust probabilistic method optimized for large-scale genotype data [7]. The supervised implementation of SCOPE uses alternating least squares to estimate individual admixture according to $\widehat{F} = PQ$, where $\widehat{F}$ is the matrix containing individual allele frequencies, $P$ are the provided ancestral allele frequencies, and $Q$ is the individual admixture estimate [7]. $\widehat{F}$ is computed through latent subspace estimation [7]. This allows single-step estimation of admixture based on provided minor allele frequencies.

SCOPE has been used to infer ancestry in human datasets [13–15]. However, there tend to be far fewer defined subpopulations in humans (usually between 5 and 6 corresponding to geography/region) compared to dogs, which are characterized by hundreds of breeds. This study uses SCOPE to predict admixture in dog breeds based on whole genome sequence data from a reference population of 65 breeds (349 individual dogs). We developed a breed-specific and global method to select about 800,000 breed-informative single nucleotide polymorphisms (SNP). We tested our approach using synthetically-admixed subjects and real samples of known breed composition. We find that we are able to accurately predict admixture even at very low sequencing depths (<1×).

## RESULTS

### Construction of reference breed population

We sought to define the genetic structure of a set of reference dog breeds, which were chosen based on the availability of high coverage whole genome sequencing data in the SRA. A total of 349 samples spanning 65 breeds were used (Supplementary Table 1).

The reference population was constructed by aligning whole genome sequence Fastq files collected from SRA and aligned to the CanFam4 dog genome. Variants in each sample were called using mpileup and variant files were merged using bcftools (see Methods). This procedure generated a set of 28067497 variants. Variants whose minor allele frequencies were below 0.01 or genotyping rate below 20% were removed with PLINK's -maf and -geno commands, respectively, then linkage disequilibrium (LD) pruned with a 250 kb window size, 50 variant window size, and $r^2$ threshold of 0.8, leaving a reference size of 5481445 SNPs. This formed the neutral, linkage disequilibrium pruned reference. These variants were further filtered to identify breed informative SNPs using plink -Fst. Breed-specific SNPs were chosen by modifying the cluster file to identify variants with high Fst between that breed and all other breeds combined. We tested different selection criteria with the top



10,000, 2500, 1000, and 500 SNPs with the highest FST chosen for each breed (an example modified cluster file is available in Supplementary Table 2). Additional SNPs were chosen by filtering the merged dataset for SNPs with an FST above 0.350 across all 65 breeds. The different FST filtering thresholds resulted in SNP reference sizes 797077, 459847, 418707, and 410339 variants respectively. To understand which set of SNPs led to optimal clustering of samples from each breed, we created UMAPs of each proposed reference and calculated the distance between each sample and its breed-specific centroid. We found that the reference with 10,000 breed-specific variants from each breed had the lowest average distance between samples and their clusters (0.055), whereas the references with 2500, 1000, and 500 breed-specific variants per breed had average distances of 0.057, 0.060, and 0.062, respectively. The neutral, linkage disequilibrium pruned reference had an average distance of 0.086. The UMAPs and distance distributions of these references are reported in the supplementary materials (Supplementary Figures 1–3 and Supplementary Tables 3–6). Because it had the lowest average distance between samples and their breed-specific centroids, we used the reference with 10,000 breed-specific SNPs when estimating admixture in synthetic and known composition mixed breed samples.

The distributions of the FST values of the 10,000 breed-specific SNPs are shown in Figure 1. Boerboels had the lowest FST values, while several breeds, such as Basenji, had estimates near 1. To visualize allele frequencies across breeds, we performed kmeans clustering of the SNP minor allele frequencies. The results of the breed-clustered heatmap are shown in Figure 2. The clustering of West Highland White Terriers, Cairn Terriers, and Scottish Terriers has been observed in prior studies and is reflected in Figure 2 [2, 16]. Interestingly, some breeds showed nearly identical allele frequency distributions, indicating no breed-specific SNP clusters. The relatively low MAF's observed in the German Shepherd column likely stem from the fact that the CanFam4 assembly is itself a German Shepherd. We observed that on average, the German Shepherd samples used during reference construction tended to have fewer variants compared to other breeds (Supplementary Tables 11 and 12). The clustering of Boerboel and Sloughi may be related to the relatively low FST distributions observed in Figure 1.

We next sought to understand the degree to which individuals of the same breed tend to cluster together. This was accomplished using UMAP dimension reduction of the genetic relationship IBS matrix (produced using PLINK 1.9's –make-rel command) of the FST-filtered reference population (797077 SNPs), the results of which are shown in Figure 3. Breed clusters are labeled. The resulting map shows that most individuals from the same breed tend to cluster together. We also note that certain groups of breeds tend to group together, largely by geographic origin. For example Sloughi, Basenji, and Saluki samples are found near each other on the UMAP. This group is largely Middle Eastern and African in origin. A cluster of Asian breeds is also visible, as represented by the close grouping of Shih Tzu, Lhasa Apso, Tibetan Mastiff, Shiba Inu, Chinese Sharpei, Thai Ridgeback, Kishu, and Chow Chow clusters. West Highland White Terrier, Cairn Terriers, and Scottish Terriers also tend to group together in line with previously reported relationships [2, 16]. Many of the remaining breeds are of European origin and form a large cluster. We also recapitulate the clustering of Labrador and Golden Retrievers observed in Figure 2 in the UMAP. By and large these findings were consistent with previously reported relationships [2, 16]. Many of these clusters were also reproduced when creating a neighbor-joining tree on the allele count Hamming distance matrix produced by the PLINK –distance square command

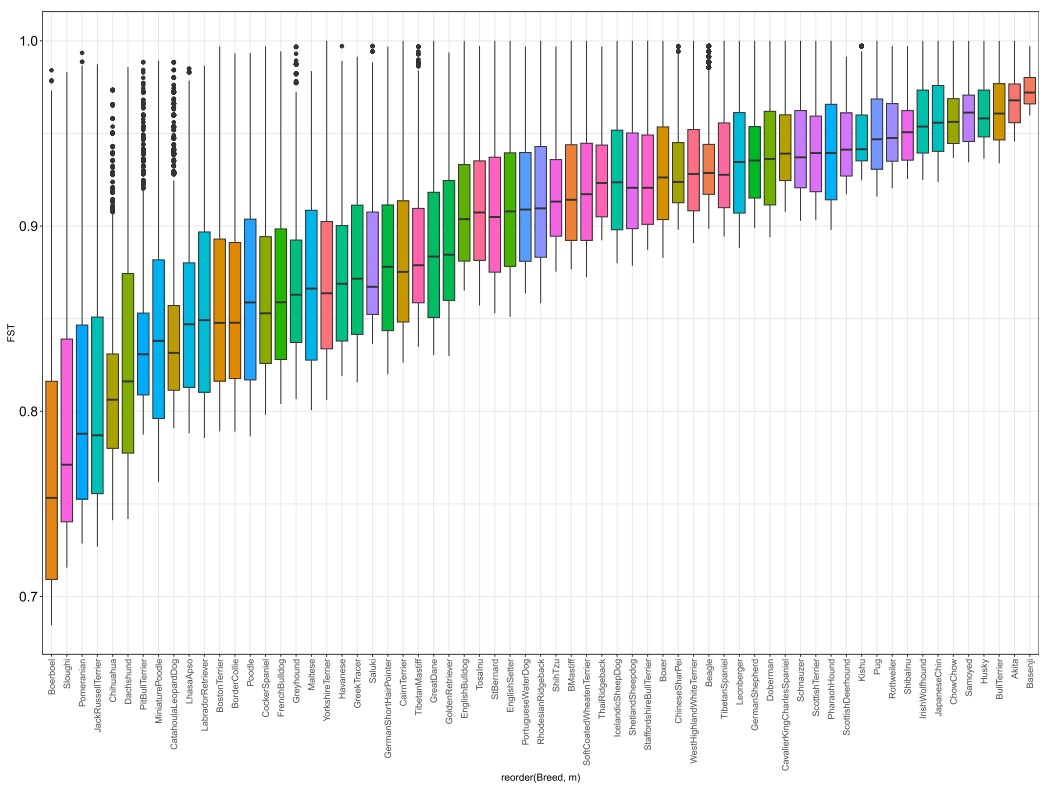

**Figure 1.** Distributions of the 10,000 SNPs with the highest FSTs collected using the breed-specific FST calculation method. Substantial variability is visible, with Boerboels and Sloughis having the smallest values, while several breeds have many SNPs with FSTs close to 1.

(Supplementary Figure 7). This neighbor joining tree was constructed to verify clustering among samples, rather than ancestry between different breeds. We also show a PCA plot, which was truncated to exclude German Shepherds, which were distant from the other breeds (Supplementary Figure 6). This is likely due to the fact that the CanFam4 reference sequence used for alignment was from a German Shepherd.

## Admixture inference

Next, we sought to estimate genetic admixture in individual dogs. We used SCOPE in a supervised mode with the allele frequencies of each SNP in each breed. We first estimated the breed admixture of each of the samples contained within the reference population, which we expect to be mostly pure breeds. The results are shown in Figure 4. Most breeds were well estimated. However, there were some exceptions, with the lowest proportion of the correct breed observed among Catahoula Leopard Dogs, Greek Tracers, and Pit Bull Terriers (Figure 4). This may be explained by the observation that Catahoula Leopard Dogs and Greek Tracers lie close to the center of the large cluster observed in the UMAP (Figure 3), which suggests difficulty in distinguishing them from other breeds. Pit Bull Terriers have relatively low breed-specific FST distributions (Figure 1), which may result in difficulty distinguishing them from other breeds. Standard deviations were bootstrapped by



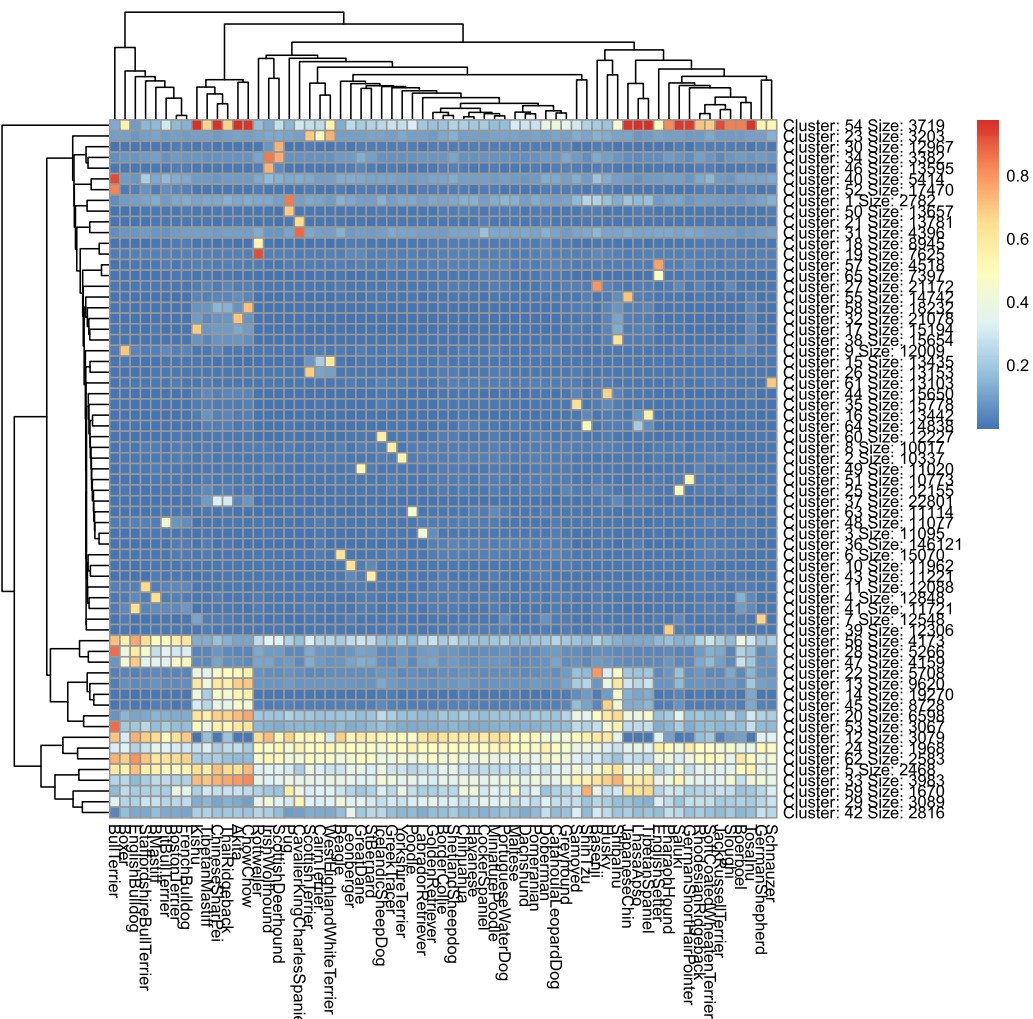

**Figure 2.** K-means clustering of breed informative SNPs (approximately 800,000 total). Clustering was performed on minor allele frequency values using the pheatmap package. Each breed shows a unique combination of minor allele frequencies, indicating successful clustering. Seed was set to 7 ahead of clustering.

randomly removing six samples from the reference population and repeating SCOPE analysis.

We next generated fifty synthetically-admixed samples from the neutral, linkage disequilibrium pruned reference with haptools' simgenotype command to measure how accurately we can infer ancestry in mixed subjects. Synthetic samples were created by randomly selecting individuals and proportions from the reference population. Results of this analysis are shown in Figure 5, with RMSE's and Pearson correlations in Supplementary Table 7. The resulting breed estimates are generally accurate, with nearly all correlation coefficients above 0.8. Four samples were below this threshold: a CatahoulaLeopardDog/BullTerrier (60.71%/39.29%), ShetlandSheepdog/TibetanSpaniel (80.19%/19.81%), a Havanese/TibetanMastiff (79.16%/20.84%), and a GreekTracer/ShihTzu (49.53%/50.47%) mix. This was not surprising as previously we noted that purebred Greek Tracer and Catahoula Leopard Dogs samples were not well estimated. Though the

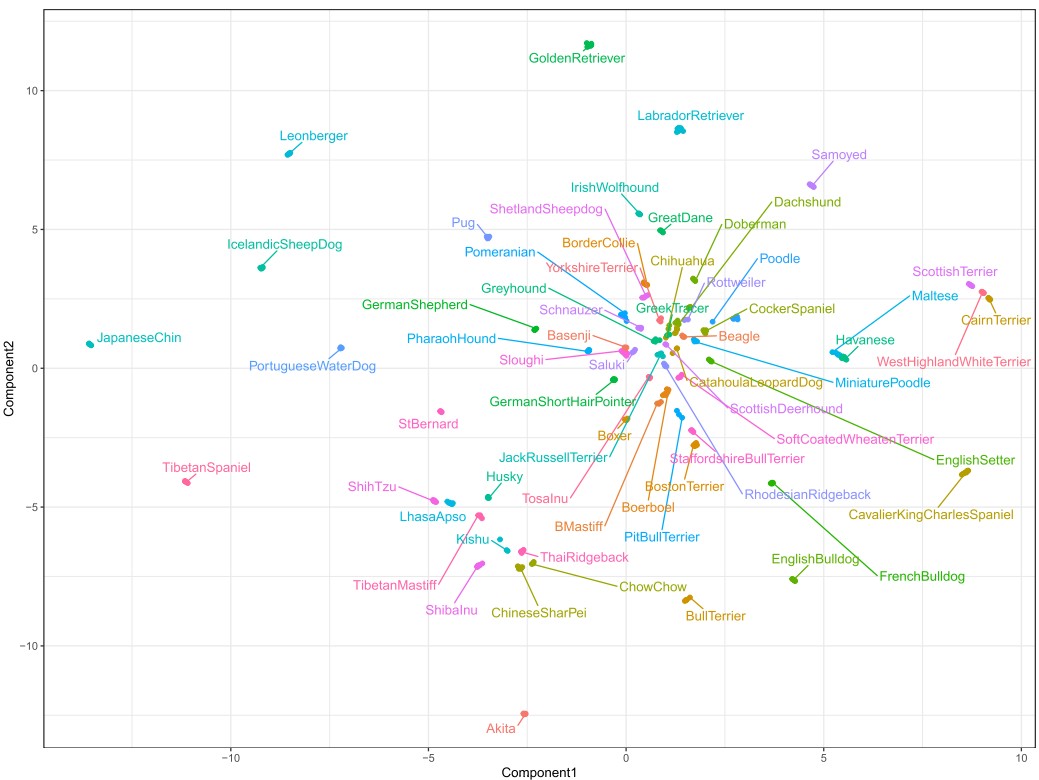

**Figure 3.** UMAP plot of the genetic relationship IBS matrix derived from the reference population with PLINK, colored by breed. Breeds tend to cluster by geography and are in line with previously reported relationships. Seed was set to 100.

correlation coefficients of the ShetlandSheepdog/TibetanSpaniel and Havanese/TibetanMastiff were relatively low, only the correct breeds were estimated, albeit in the wrong proportions.

To verify that SCOPE was able to predict breeds on samples outside the reference population, we tested mixed samples from a previous study [17] whose admixture had been assessed with third party tests. These lower coverage samples have about 2.5× depth on average and had been assessed to be 42.2% Miniature Poodle/33.9% Poodle/9.1% Labrador Retriever/14.8% Cocker Spaniel (Labradoodle #1), 34.9% Miniature Poodle/40.2% Poodle/10.4% Labrador Retriever/14.5% Cocker Spaniel (Labradoodle #2), and 43% Pomeranian/40% Chihuahua/17% Icelandic Sheepdog samples (PomChi), by third party admixture tests, respectively. In order to quantify the optimal coverage depth needed to achieve high accuracy, we downsampled the three samples. Downsampling was carried out using the sample command from seqtk (https://github.com/lh3/seqtk). We downsampled to 75%, 50%, 25%, 10%, and 5% of its original number of reads and estimated the admixture for each. For these samples, genotyping occurred at the positions of the neutral, LD pruned set of variants, the bcf.gz converted to PLINK, the intersection of SNPs between the genotyped variants and the variants in the reference population found, the sample and reference PLINKs merged using the PLINK 1.9 bmerge command, common variants extracted, and a new set of global SNPs calculated using an FST threshold of 0.350. This was



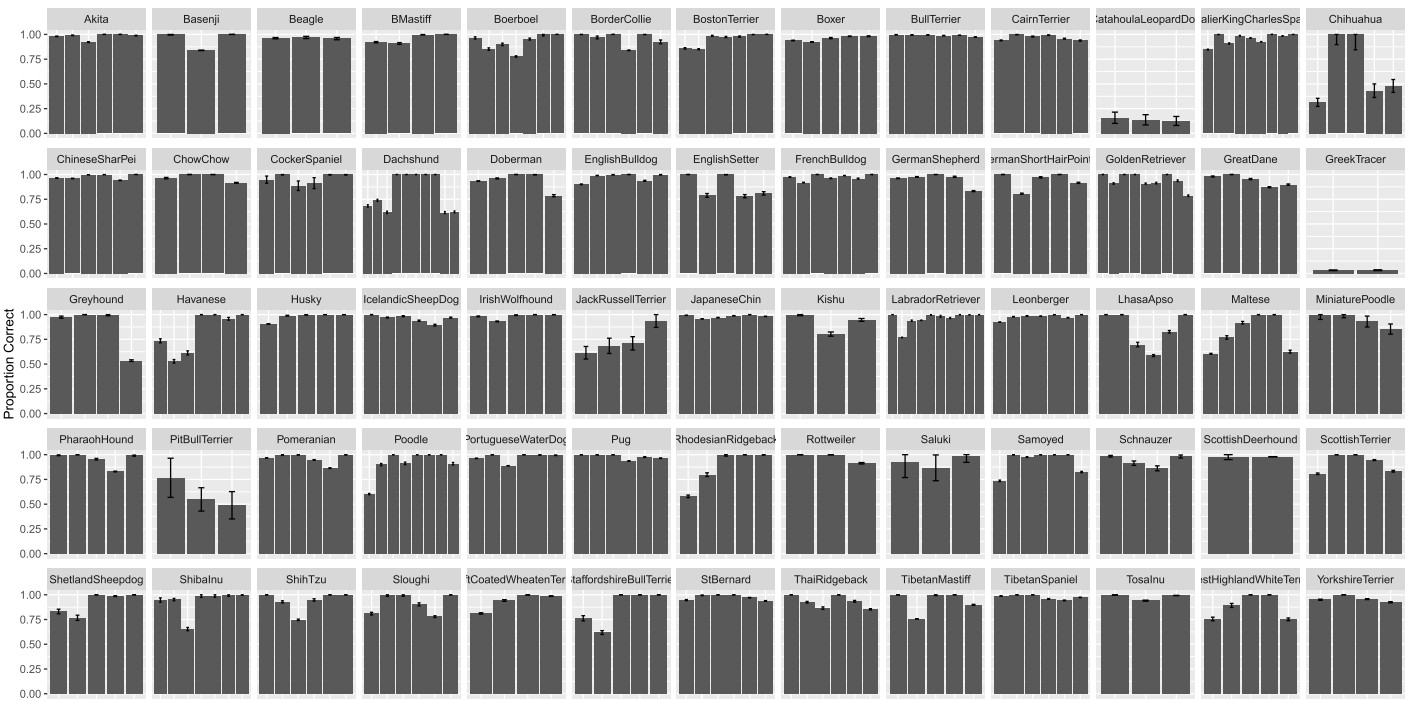

**Figure 4.** Recovery of reference breed samples with SCOPE, with the first sample of each breed labeled with the breed. The overwhelming majority of breeds are recovered accurately. Exceptions include Catahoula Leopard Dogs, Greek Tracers, and Pit Bull Terriers. Black bars represent +/– 1 standard deviation using bootstrapping, in which 6 samples were removed from the reference population and SCOPE analysis performed (100 total iterations).

concatenated with the breed-specific SNPs (10,000 variants per breed) previously calculated and extracted from the merged sampleset and SCOPE analysis performed. The results of this analysis are shown in Figure 6 and RMSE's and Pearson correlations in Supplementary Table 8. We did not observe significant decreases in accuracy when downsampling, which suggests that our filtering technique is robust at low sequencing depths. The average correlation coefficient across all samples and depths was 0.85 (0.77 for PomChi, 0.85 for Labradoodle #1, and 0.95 for Labradoodle #2).

## DISCUSSION

We present a study of the genetic structure across dog breeds as well as an approach to estimate admixture from whole genome sequence data. After assembling a reference population of 349 dogs spanning 65 breeds, we identified SNPs across this population that have high FST values when comparing a single breed to the rest of the breeds, as well as SNPs that have high FST when comparing across all breeds. Dogs of the same breed were shown to cluster together. We next used SCOPE to infer admixture and found that with a few exceptions the breed of most pure bred dogs is well estimated. Finally, we show that this approach can recover the breed mixture estimates of mixed dogs, and that the quality of the estimates depend weakly on sequencing coverage.

We find that including a greater number of breed-specific SNPs improves the ability to cluster same-breed samples. Based on this result, we used a 797077 SNP reference panel composed of 10,000 breed-specific SNPs from each dog breed and a global set of SNPs derived from filtering the neutral reference at an FST threshold of 0.35. The distributions of



**Figure 5.** Synthetic data recovery. Synthetic samples were generated by randomly selecting SNPs from two random samples contained within the reference population. Recovery was largely successful, indicating that most tested samples could be profiled. Samples are sorted in order of increasing root mean square error.

breed-specific FST values, as well as UMAP clustering reflected which breeds were more difficult to estimate during admixture analyses. Among breeds close to the center of the large, European cluster observed in the UMAP (Figure 3), Greek Tracer and Catahoula Leopard Dogs were difficult to correctly predict in the original reference population (Figure 4) and during synthetic sample analysis (Figure 5). Challenges estimating the admixture of Pit Bull Terriers were observed in the admixture analysis of the reference population, but not during synthetic sample analysis (both samples containing Pit Bull Terrier admixture had correlation coefficients above 0.9). Interestingly, Catahoula Leopard Dog was often overrepresented during the reference recovery experiment. Challenges in estimating Pit Bull Terrier ancestry/admixture may also be due to the definition of the breed being based on phenotype rather than pedigree, as there is no established DNA signature for Pit Bull Terriers [18]. Comparisons of DNA-based and visual estimations of Pit

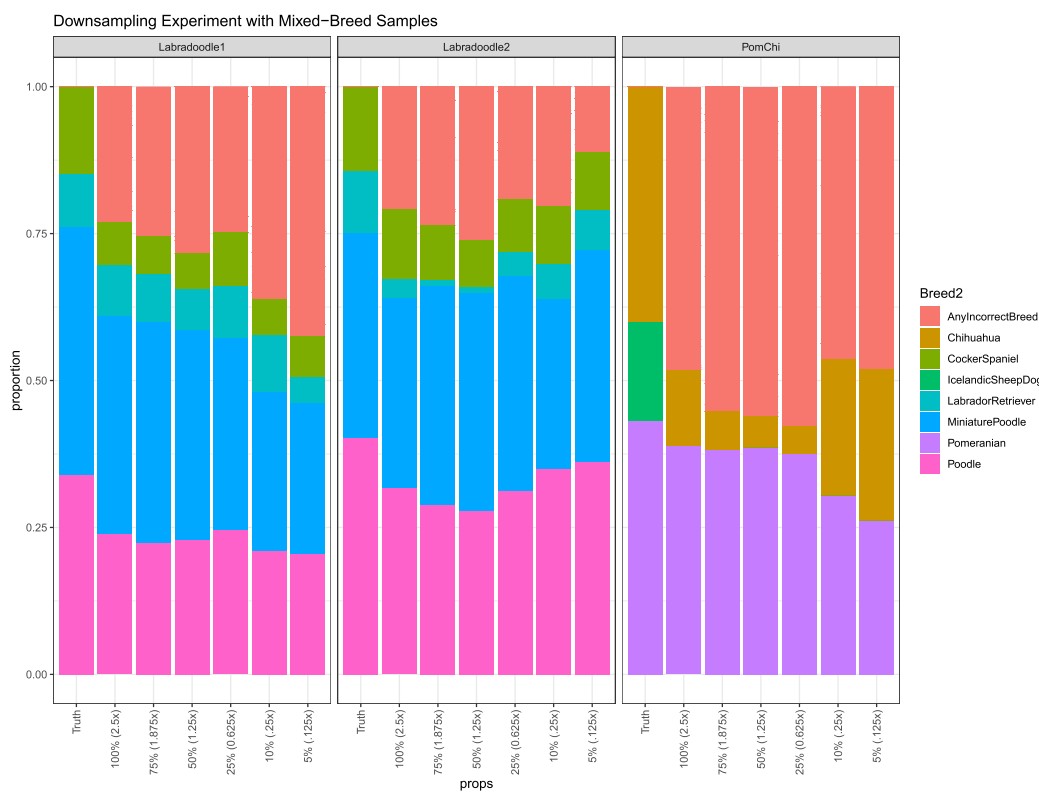

**Figure 6.** Detailed downsampling experiment of mixed breed dogs. Results show relatively consistent admixture estimations across a range of sequencing depths.

Bull Breeds have had poor consistency, which may suggest substantial outbreeding and that there could be several genotypes associated with the Pit Bull Terrier breed [18].

We also sought to determine the genome coverage levels that are needed for robust estimation of admixture. For three samples not contained in the reference population, we found that SCOPE admixture estimates remained stable across different read counts/sequencing depths.

Future efforts could seek to improve the accuracy of different admixture inference tools across coverage levels. Maximum likelihood approaches, such as those employed by fastNGSadmix, present themselves as potential ways of approaching this challenge, though it has presently been used for human populations, which tend to have fewer distinct groups compared to dogs [19].

Although we focused on 65 breeds, there are several hundred breeds that are recognized by international institutions. Thus future studies should expand the number of samples and breeds within the reference population and examine how the approaches presented here scale with increased numbers of breeds. Further investigation will be needed to confirm whether the addition of other individuals from these breeds could improve the identification of breed informative SNPs and accurate estimate of their allele frequencies. We also recognize that the number of third party validated samples is relatively low and a greater number will be needed to verify the results of this study.

## LIMITATIONS

The findings of this study are limited in several ways. Some breeds are underrepresented in the reference panel due to limited data availability. The low numbers of Greek Tracer, Catahoula Leopard Dog, and Pitbull Terrier samples may have contributed to difficulties estimating ancestry. Additionally, the underrepresentation of other breeds (Kishu, Sloughi, Rottweiler, etc.) may lead to difficulties capturing all of the genetic variability that exists in these breeds, which significantly limits the generalizability of our findings. The number of externally validated mixed breed samples also limits our ability to assess the effectiveness of SCOPE on admixed canine samples. The number of breeds represented in these external subjects is relatively small (7 as opposed to the 65 in the reference population), which limits our ability to make conclusions about SCOPE's ability to estimate admixture in highly admixed individuals. For these reasons, our findings should largely be considered preliminary and future work should aim to expand the number of samples within each breed, as well as the number of externally validated samples tested. An expanded reference population, with substantially more samples per breed, as well additional breeds, will be necessary to confirm our findings.

## METHODS

### Reference population construction and analysis
#### *Reference population*

Fastq files retrieved from the Sequence Read Archive (SRA, full list of accession numbers and breeds are reported in Supplementary Table 1) were aligned to the CanFam4 assembly with the bwa-mem2 (version 2.2.1) mem command [20, 21]. Variants were called at each chromosome with bcftools version 1.18 mpileup and call (options -mv and -V indels were used in the calling step), and concatenated with the bcftools concat command to create individual sample bcf files. To ensure high quality variants in the reference, only SNPs with a QUAL score above 20 were kept. These quality controlled samples were then merged with bcftools merge -0. This merged reference was converted to binary file format with PLINK 1.9 (www.cog-genomics.org/plink/1.9/). This resulted in an initial reference size of 28067497 variants. Rare variants (MAF < 0.01) and variants with a genotyping rate below 20% removed with PLINK's -maf and -geno filters [22, 23]. This resulted in a reference size of 14383121 variants. Finally, linkage disequilibrium pruning was performed with PLINK's –indep-pairwise command with a 250 kb window size, 50 variant window size, and $r^2$ threshold of 0.8. This shrank the reference size to 5481445. We assumed all samples in the reference population to be pure breeds and performed UMAP clustering to verify assignments. We identified one sample (SRR35396290) which had been mislabeled as a Saluki, but clustered extremely close to other Samoyed samples. This UMAP is available as Supplementary Figure 4 and the corrected version as Supplementary Figure 5. Distances between samples and their breed-specific centroid are available in Supplementary Table 9 (corrected version is Supplementary Table 10).

To identify breed-informative SNPs, we filtered the initial SNP panel in two ways. The first approach, which we define as global filtering, used the PLINK -FST output to select SNPs with an FST above 0.350 when using the cluster file with all breeds (Supplementary Table 1). The second method, referred to as breed-specific filtering, involved collecting SNPs with PLINK 1.9's FST command with modified cluster files where only the breed of interest was labelled (and all other breeds labelled as not the breed of interest, as shown in the

example cluster file in Supplementary Table 2). The top 10,000, 2500, 1000, and 500 SNPs for each breed were collected and combined with the globally filtered SNPs, duplicates removed, and extracted from the initial, unfiltered reference with PLINK 1.9's -extract command. To determine which SNP panel best clustered samples, the distances between each sample and their breed-specific centroid were calculated.

Minor allele frequencies (MAF) of all variants in the reference's stratified frequency file (output of PLINK's –freq command) were k-means clustered with the R package pheatmap version 1.0.13 with 7 clusters [24]. Their distributions were also plotted with ggplot2 (version 3.5.2) R version 4.3.3 was used [25].

UMAP was used to visualize intersample distances computed using the genetic relationship IBS matrix calculated with PLINK's -make-rel square command. This matrix was reduced to two dimensions using uwot's (version 0.2.3) umap2 command [26]. Plotting was performed using ggplot2 and labeling with ggrepel (version 0.9.6) [27, 28]. Seed was set to 100 ahead of analysis.

### Generation of synthetic admixed individuals

Synthetic datasets were generated using the haptools simgenotype command on a vcf of the neutral, linkage disequilibrium pruned dataset [29]. Random breeds and proportions were selected using the synth_helper.sh script. This script randomly chooses two breeds and random proportions using the shuf command and assembles them to create the model file needed by haptools simgenotype. This approach preserves haplotyping information.

### SCOPE analyses

The supervised mode of SCOPE was used with the cluster stratified frequency file generated from the PLINK -freq command using the fully defined cluster file (Supplementary Table 1) [7]. Plotting of the group frequency matrix was performed with ggplot2. We also plotted the proportion of the correct breeds predicted by SCOPE. To generate standard deviations, we performed 100 bootstrapping iterations in which 6 random samples (representing about 2% of the reference population) were removed from the reference followed by admixture inference. For the synthetic data analyses, bcf files were merged with the reference variant file, converted to PLINK binary, then the previously calculated pool of SNPs extracted before conducting the SCOPE analysis described above. For real samples of known admixture, genotyping (variant calling) occurred at the neutral, LD pruned set of variants, converted to PLINK, the intersection of SNPs between the genotyped variants and the variants in the reference population found using the -comm command, the sample and reference PLINKs merged using the PLINK 1.9 bmerge command, common variants extracted, then a new set of global SNPs calculated using an FST threshold of 0.350. This pool of global SNPs was concatenated with the breed-specific SNPs (10,000 variants per breed) previously calculated and extracted from the merged sampleset and SCOPE analysis performed.

### Phylogenetic tree generation

The genetic distance matrix generated with PLINK command (–distance square) was converted into a neighbor-joining phylogenetic tree with the R package ape (version 5.8.1) commands nj() [30]. No root was explicitly set during tree construction.

## AVAILABILITY OF SOURCE CODE AND REQUIREMENTS

Project Name: Ancestry Inference
Project Home Page: https://github.com/gKislik/AncestryInference
Operating System: Platform Independent (analysis code), Linux (SCOPE [7])
Programming Language: R, C++ (SCOPE)
Other Requirements: R 4.3.3 or higher
License: MIT

## DATA AVAILABILITY

A list of individual NCBI SRA accession numbers is available in Supplementary Table 1, and all the Supplementary Tables and Figures are deposited in Zenodo [31]. These The processed reference population (filt-bcftools-ref-redone.imputed.filtered.bcf.gz) is available in the Github repository [32].

## DECLARATIONS

### Ethics approval and consent to participate

The authors declare that ethical approval was not required for this type of research.

### Competing interests

MP is affiliated with ProsperK9, which developed a direct to consumer test for dog ancestry.

### Authors' contributions

GK wrote portions of the manuscript, curated data, and wrote the analysis code. GTM wrote portions of the analysis code. LR assisted in data curation for the three non-reference population samples. VS and GC helped with data curation. MP participated in writing the manuscript, conceptualization, and helped supervise the work.

### Funding

Not applicable.

### Acknowledgements

Not applicable.

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
