## [Editor Report]

Editor’s AssessmentThe peer review of this manuscript has been completed, and it is ready for formal acceptance.Editor’s AssessmentThe peer review of this manuscript has been completed, and it is ready for formal acceptance.

---

## [Reviewer Report]

Indicate in the comments box below whether you are happy with the changes made or if the manuscript is unacceptable.Comments on revised manuscriptI feel that insufficient changes were made to the set up of the study to satisfy the comments from both reviewers. The authors changed their data processing sufficiently to address both reviewers comments on missingness and origin of the dataset. Outside of this change I found many areas to still be lacking nuance needed to justify the accuracy of SCOPE on dogs. The small sample size of true mixed dogs, two of which were the same mix, has not convinced me that the method is robust. This is not sufficient proof that the method works. Furthermore the lack of thorough investigation of the impact of unbalanced reference panel has not satisfied me for the application of SCOPE on dogs. The authors claim that they have addressed this limitation, but this is only done in the most vague sense. Their justification to reviewers that they are looking for admixture and not ancestry does not seem appropriate. I understand that SCOPE is looking for admixture, but ancestry need to be taken into account for this to be reliable. What happens if a mixed breed dog with 25% Sloughi, 25% Saluki, 25% Siberian Husky, 25% German Pointer ancestry is tested? Three quarters of the ancestry of that dog would be highly underrepresented in the reference panel. The mixing of dogs in the present day is not restricted to the breed's geographical origin. The example I just gave could be found in an Alaskan Husky and would not be uncommon.

---

## [Reviewer Report]

Upload additional filesTRR-202509-01R02/stage_files/TRR-202509-01/Review MS/Inference_of_admixture_in_dogs_from_whole_genome_sequences.pdfReviewer name and names of any other individual's who aided in reviewerTracy SmithDo you understand and agree to our policy of having open and named reviews, and having your review included with the published manuscript. (If no, please inform the editor that you cannot review this manuscript.)YesIs the language of sufficient quality?YesPlease add additional comments on language quality to clarify if neededIs there a clear statement of need explaining what problems the software is designed to solve and who the target audience is? YesAdditional CommentsIs the source code available, and has an appropriate Open Source Initiative license <a href="https://opensource.org/licenses" target="_blank">(https://opensource.org/licenses)</a> been assigned to the code?YesAdditional CommentsAs Open Source Software are there guidelines on how to contribute, report issues or seek support on the code?YesAdditional CommentsIs the code executable?Unable to testAdditional CommentsIs installation/deployment sufficiently outlined in the paper and documentation, and does it proceed as outlined?Unable to testAdditional Commentsscripts are provided, but I have not tested them.Is the documentation provided clear and user friendly?YesAdditional CommentsIs there enough clear information in the documentation to install, run and test this tool, including information on where to seek help if required?YesAdditional CommentsIs there a clearly-stated list of dependencies, and is the core functionality of the software documented to a satisfactory level?NoAdditional Commentswould like to see full plink code too as well as R dependenciesHave any claims of performance been sufficiently tested and compared to other commonly-used packages? NoAdditional CommentsIs test data available, either included with the submission or openly available via cited third party sources (e.g. accession numbers, data DOIs)?YesAdditional CommentsAre there (ideally real world) examples demonstrating use of the software? YesAdditional CommentsIs automated testing used or are there manual steps described so that the functionality of the software can be verified?Additional CommentsAny Additional Overall Comments to the AuthorRecommendationMajor Revisions

---

## [Reviewer Report]

Reviewer name and names of any other individual's who aided in reviewerTatiana FeuerbornDo you understand and agree to our policy of having open and named reviews, and having your review included with the published manuscript. (If no, please inform the editor that you cannot review this manuscript.)YesIs the language of sufficient quality?YesPlease add additional comments on language quality to clarify if neededIs there a clear statement of need explaining what problems the software is designed to solve and who the target audience is? YesAdditional CommentsIs the source code available, and has an appropriate Open Source Initiative license <a href="https://opensource.org/licenses" target="_blank">(https://opensource.org/licenses)</a> been assigned to the code?YesAdditional CommentsAs Open Source Software are there guidelines on how to contribute, report issues or seek support on the code?YesAdditional CommentsIs the code executable?Unable to testAdditional CommentsIs installation/deployment sufficiently outlined in the paper and documentation, and does it proceed as outlined?YesAdditional CommentsIs the documentation provided clear and user friendly?YesAdditional CommentsIs there enough clear information in the documentation to install, run and test this tool, including information on where to seek help if required?Additional CommentsIs there a clearly-stated list of dependencies, and is the core functionality of the software documented to a satisfactory level?YesAdditional CommentsHave any claims of performance been sufficiently tested and compared to other commonly-used packages? NoAdditional CommentsThere is no comparison of the results to other commonly-used packages. The only comparison is to published data that the authors accept as accurate.Is test data available, either included with the submission or openly available via cited third party sources (e.g. accession numbers, data DOIs)?YesAdditional CommentsAre there (ideally real world) examples demonstrating use of the software? YesAdditional CommentsThe real world example is an insufficient number of samples to be indicative of the accuracy.Is automated testing used or are there manual steps described so that the functionality of the software can be verified?Additional CommentsAny Additional Overall Comments to the AuthorThe premise of the paper could be an interesting test of the use of the SCOPE software on dogs. I can appreciate the idea of the manuscript and the profile journal selected for the submission, but even for the journal the intentions of the journal don't appear to align fully with the way the testing of the method was carried out. Additionally, the dataset of dog breeds is insufficient to be informative. This is particularly true of the number of mixed dogs tested and the down-sampling. Furthermore, any interpretation of the results lacks the observation of these limitations and the nuance of the geographical bias of the dataset. Reviewer comments: “Bergstrom et al. 2012” wrong citation “Global ancestry, inferred by tools such as SCOPE and ADMIXTURE (Alexander et al. 2009), attempts to infer the proportions of an individual’s genome that belong to an ancestral breed or group.” Citation for SCOPE missing If studies such as Parker et al 2017 have used 160 breeds and the authors have mentioned the numerous subpopulations of dogs, why did the authors choose to use such a small number of breeds for their study? Why were the top 2500 SNPs used? Why not 1000 or 10000, etc? Testing the number that are needed would be very informative. Figure 1, I would recommend sorting the breeds by value so that the results can be interpreted more easily. “We also note that certain groups of breeds tend to group together. For example Samoyed, Basenji, and Husky samples are found near each other on the UMAP. This group has been shown to represent ancient breeds (Larson et al. 2012, Pickrell and Pritchard 2012, Wojcik and Powierza 2021).” There are other explanations for this pattern, almost all of the other breeds examined are breeds of European origin, there is very little representation of non-European ancestry within the small sample size of dog breeds included in the study. “Despite this the more distant relationships between breeds differ from some of the previous studies, as these may be more difficult to define using our markers.” If this is the case and could be influencing results, it would be relevant to mention which breeds these are. Why is SNP chip data rather than whole genome sequencing being used for the study? This should be clearly established, any explanation for this choice is completely absent. Is it because SCOPE can only handle a small number of sites? Is it because of the availability of the dataset? If so, note my previous concern with the small sample size, despite the public availability of much larger datasets. Is there another reason? Figure 5, The size of the legend versus the figure itself is very unbalanced, and I would recommend making a clearer delineation of the breeds, it is unclear where one breed ends and the next begins. The size of the figure is also difficult to see the individuals with more than one bar colour. A continuous colour scheme is also probably the wrong choice for the plot, the already difficult delineation of the breeds is nearly impossible, given that the breeds are sorted alphabetically I know the colour choice is purely incidental thus making the continuous palate even more inappropriate. Figure 7, most of my comments on Figure 5 also apply to Figure 7. The colour choices make it very difficult to see how many segments are present in each bar. Also an indicator of which simulated individuals were determined to be successful versus unsuccessful would be helpful. For example the rightmost four bars look fairly unsuccessful to me as they are all missing a component in the estimate that was present in the truth. Using three mixed dogs seems like a very small number of samples to test the accuracy of the tool on real datasets. Downsampling only one individual to test the impact of coverage is likely not representative of the impact of low coverage across all breed compositions. A larger number of individuals downsampled would be more informative for the accuracy of the results. In the discussion in page 11, many old citations are used to back up the interpretation of the close relationship of Siberian Huskies, Basenjis, and other non-European breeds. No mention is made of the geographical bias of the dataset as a reason behind this as I previously mentioned. General comments: There are frequent issues with a lack of spaces ‘ ‘ between parentheses and neighbouring words and punctuation. A different colour palette should be used for the figures. It is very difficult to determine the breed due to the poor colour choice used throughout the manuscript. Inconsistent citation styles are used, eg. “Similarly, prior maximum likelihood estimation based techniques have suggested that Huskies and Samoyeds are both ancient breeds and related to Basenji (23, 29).”RecommendationMajor Revisions